# Calcification of the Atlanto-Occipital Ligament (Ponticulus Posticus) in Orthodontic Patients: A Retrospective Study

**DOI:** 10.3390/healthcare10071234

**Published:** 2022-07-02

**Authors:** Daniela Di Venere, Alessandra Laforgia, Daniela Azzollini, Giuseppe Barile, Andrea De Giacomo, Alessio Danilo Inchingolo, Biagio Rapone, Saverio Capodiferro, Rada Kazakova, Massimo Corsalini

**Affiliations:** 1Interdisciplinary Department of Medicine, University of Bari ‘Aldo Moro’, 70121 Bari, Italy; daniela.divenere@uniba.it (D.D.V.); alessandra.laforgia@uniba.it (A.L.); daniela.azzollini93@gmail.com (D.A.); ad.inchingolo@libero.it (A.D.I.); biagiorapone79@gmail.com (B.R.); capodiferro.saverio@gmail.com (S.C.); 2Department of Basic Medical Sciences, Neurosciences and Sense Organs, University of Bari ‘Aldo Moro’, 70124 Bari, Italy; andrea.degiacomo@uniba.it; 3Department of Prosthetic Dentistry, Faculty of Dental Medicine, Medical University—Plovdiv, 4002 Plovdiv, Bulgaria; rada.kazakova@mu-plovdiv.bg

**Keywords:** normal anatomy, ponticulus posticus, lateral cephalometric radiography, skeletal malocclusion, dental anomalies

## Abstract

Background: Ponticlus Posticus (PP) is a rare anomaly of the first cervical vertebra easily identifiable in lateral cephalometric radiograph and typically required for orthodontic diagnosis. The aim of this study is to evaluate the PP prevalence in lateral Cephalograms in a cohort of orthodontic patients treated at the Dental School of the University of Bari ‘Aldo Moro’, Italy, and to find possible connection between PP and other dental anomalies, as well as the patient’s cephalometric characteristics. Methods: A total of 150 panoramic radiographs and 150 lateral Cephalograms, obtained for orthodontic use only, were evaluated. No patients were referred to for congenital syndromes or disease, or a history of previously occurred maxillofacial trauma. A detailed cephalometric study was performed for each patient, and the whole cohort was divided according to the common three dental malocclusion classes. The values obtained were analyzed using the Chi-Square Test. Results: PP prevalence was 8% (12 of 150 cases), where the complete and partial forms observed 4.7% and 3.3%, respectively. Although females were more affected (9.6%), no statistically significant sex-related difference was found. Furthermore, no statistically significant relationship regarding age or dental anomalies (dental agenesis/palatal impacted canines) among the groups was observed either. Cephalometric analyses revealed that half of the subjects with PP were sagittal skeletal class I and had vertical hyper-divergence. Conclusions: PP is a frequent anatomical variation of the cervical vertebra, apparently unrelated to the skeletal malocclusion type or dental anomalies. The current study needs to further confirm the congenital hypothesis of PP’s origin already reported in literature.

## 1. Introduction

The normal anatomy of the first cervical vertebra (C1) may exhibit several variations, the most common among them is the calcification of the atlanto-occipital ligament, also known as Ponticulus Posticus (PP) [1]. The PP is a bony bridge located over the vertebral groove, arising from the posterior portion of the superior articular process of the atlas and the posterolateral portion of the superior margin of the posterior arch of the atlas. It could completely or partially encompass the vertebral artery and the first cervical nerve [2]. PP may present as a uni- or bi-lateral form, and, depending on its extent of calcification, is generally distinguished as partial or complete [3]. As for the radiological appearance, PP is clearly identifiable in an X-ray of the cranio-vertebral junction in lateral projection or in a lateral Cephalogram of the skull, the latter remaining commonly used for orthodontic diagnosis [4]. However, regular radiograms cannot distinguish a bilateral from a unilateral form [5]. Different radiological techniques may be required to obtain adequate clinical imaging, and are usually case-dependent [6]. The bony bridge, generally known as PP, occurs in the literature under various names; i.e., Kimmerle’s anomaly, arcuate foramen or Sagittal foramen [2,3]. PP etiology is still unknown but several hypotheses have been postulated, such as: genetic nature/heredity [7]; a congenital malformation [5,8,9]; acquired calcification of the atlanto-occipital ligament due to different factors such as aging, artery pulsation, embryonal residues activation with osteogenic activity, or external mechanical factors, including poor posture or trauma [2]. Data regarding PP prevalence are highly dependent on the geographic region and ethnicity of the examined subjects and it is generally reported to be most common in the Western population, with a prevalence rate between 5.1% to 37.8% [5]. Moreover, data on gender distribution are also quite discordant in the literature; in 2017, a study on orthodontic patients from North Italy demonstrated a prevalence rate of 7.7% of the complete form of PP, and 9.0% of the incomplete one, occurring in 8.8% and 11.0% in males, and 6.9% and 7.7% in females, respectively [10]. PP patients are frequently unaware because its occurrence is asymptomatic in the majority of the cases. However, some rotational or extensive neck movements may cause a temporary compression of the vascular and nervous structures of the bony bridge, resulting in different degrees of complaints. Mild symptoms may range from neck pain, radiating to the limbs, to dizziness and migraine with and without aura [3], while severe complaints generally include loss of consciousness, vertebrobasilar insufficiency and cerebellar infarction [3,11]. In addition, serious complications could occur during surgery, predominantly due to failure in PP recognition; e.g., artery perforation during surgical screws insertion for atlanto-cervical stabilization, or vertebral artery occlusion during surgeries involving neck hyperextension, with possible cerebellar infarction [5,8,11,12]. PP is also a predisposition to both dental and general conditions possibly compromising the general health status of patients. Therefore, many authors tried to find a possible correlations between PP and other skeletal disorders including the sella turcica bridging [13,14], or elongated styloid process [15,16]. Some authors pointed out that agenesis/impaction of the upper canines may be related to skeletal alterations, including PP [17,18,19]. Other authors found an association between cleft lip and palate (CLP) and PP, and suggested that the ‘defective’ mesenchymal development, responsible for CLP, may also cause abnormalities at sites distant from the cleft [20]. Moreover, evidence of PP in neoplastic disorders such as the Gorlin-Goltz syndrome, suggests that it could represent an early indication for a diagnosis [21,22]. The aim of the current retrospective study is to evaluate the prevalence of PP detection on lateral cephalometric X-rays in patients attending the Dental School of the University of Bari, Italy, for orthodontic purposes, demonstrating a possible association between PP and dental anomalies, and reporting the relevant cephalometric features.

## 2. Materials and Methods

This study was carried out according to the Declaration of Helsinki and approved by the Independent Ethical Committee active in the Faculty of Medical Sciences, University of Tetovo, North Macedonia (Study No. 09-154/1, dated 15 February 2022).

A retrospective observational analysis on lateral cephalometric radiographs of the skull and Orthopantomographs (OPT) was conducted for an orthodontic analysis of patients, treated at Dental School of the University of Bari ‘Aldo Moro’, Italy. All of the enrolled patients’ parents signed an informed consent at the time of the first clinical examination. Exclusion criteria included: incomplete documentation, age over 18 years, pre-existing already diagnosed syndromic or systemic disease, history of dento-facial trauma, and radiographs with poor-defined cervical spine. Data regarding sex and age at the time of radiological investigation were collected.

PP recognition was performed by direct visualization on lateral cephalometric X-rays of the skull (Figure 1 and Figure 2); each radiogram was evaluated twice by the same orthodontist and in a dubious case, a blind second opinion was required from a separate clinician. 

The presence of PP was reported as follows: absence of calcification; partial PP, when the calcification of the atlanto-occipital ligament from the lateral mass to the posterior arch of C1 was discontinuous, complete PP in cases of evident bony ring above the C1 posterior arch.

A sagittal and vertical skeletal evaluation was performed by a cephalometric analysis of the SNA angle, SNB angle, ANB angle, SnaSnp-GoGn intermaxillary angle, SN-GoGn craniomandibular angle. The antero-posterior relationship of the maxilla and mandible was classified according to the conventional skeletal class differentiation as I, II, and III, based on the width of the ANB angle (Class I: ANB between 0–4°; Class II: ANB > 4°; Class III: ANB < 0°). The vertical relationship of the maxillae was defined as meso-divergent, hypodivergent, and hyperdivergent (intermaxillary angle between 15–25°, <15°, and >25°, respectively), and as open or deep based on the width of the craniomandibular angle (<27°—deep skeletal, and >37°—open skeletal). OPTs were carefully inspected for detecting dental anomalies such as dental agenesis and canine inclusion. 

The collected data were reported in terms of mean and standard deviation, percentage, and prevalence. The prevalence in males and females was compared with the Chi-square test setting statistical significance for *p*-value < 0.05. The sample was divided into 2 age groups (4–11 years and 12–18 years); age was compared by the Chi-square test in the groups of subjects with and without PP to find possible age-related differences. The Chi-square test with Yates correction was used to compare the distribution difference of dental abnormalities in patients with and without PP. The sample was divided in skeletal malocclusion groups according to the ANB angle, and a comparison between the groups of patients presenting anomalies or not, by using the Chi-square test, was performed.

## 3. Results

Considering the inclusion and exclusion criteria, 150 patients were included in the study. A patient suffering from Incontinentia Pigmenti was excluded from the study despite having a complete form of PP. Sample characteristics are described in Table 1. 

Sex distribution was homogeneous: 45% males and 55% females. Age ranged from 4 years and 9 months to 17 years and 1 month, and the mean age was 10.3 (±2.7). The PP patients’ group’s mean age was 11.16 (±2.08) years. PP was observed in 12 patients, a prevalence of 8%. The partial form was found in 3.3%, and the complete form in 4.7% of the cases. Females presented with a higher prevalence of PP (9.6%) than males (5.9%). The complete form was more frequent in males (75%), whereas the two forms occurred equally in women. However, the analysis did not show statistically significant differences (Table 2). 

As far as frequency in the two age groups was concerned, most cases were between 4 and 11 years with a prevalence of 8.2%; in the 12–18 years range, PP was present in 7.7%. No significant age-related difference of the presence of PP was observed (Table 3). 

As for teeth number abnormalities (multiple agenesis) and eruption abnormalities (retained canines), 2% of the patients had dental agenesis (3/150; 1 male and 2 females), and among these only one presented with a complete form PP. No significant difference regarding the presence of agenesis among patient groups with and without PP was observed (Table 4).

Palatal impacted canines were exhibited by the 10.66% of the subjects with a low female prevalence (7 males and 9 females), and only three of them had PP, two—the partial form and one—the complete form in particular, yet without a statistically significant difference regarding the canine retention among the patient groups with and without PP (Table 5).

In terms of vertical cephalometric analysis, the open skeletal class was present in half of the subjects (6/12), and none of them had maxillary hypo-divergence. Sagittal analysis revealed skeletal class I as the most frequent one (6/12), followed by skeletal class II (4/12) and, ultimately, skeletal class III (2/12). The results obtained were compared by means of the cephalometric values of the group without the PP using Chi-square tests. The statistical analysis revealed no significant difference (Table 6).

## 4. Discussion

The latero-lateral cephalometric analysis of the skull is a routine radiological examination for orthodontic diagnosis and treatment planning. In such a radiograph, the cervical spine is studied to determine the degree of patient’s bone maturation [23]; PP can be easily discerned as it resembles a well-demarcated radiopaque area at the C1 vertebral artery sulcus.

PP is described as a common anatomical variant of C1 in the literature, with a variable prevalence; lower incidence is reported in India and South Korea, whilst it is higher in North America [12]; in the Western countries, the percentage is between 5.1% and 37.8% with a slight female predominance [5]. PP occurrence in Italy is reported to be 16.7% for orthodontic patients in Northern Italy, with a higher frequency of the partial form of PP [10].

Only 12 patients out of 150 from our study exhibited PP, a prevalence of 8%. In contrast to Gibelli who reported PP prevalence of 7.7% for the complete form and 9.0% for the partial one [10], the complete form in our study revealed slight predominance (4.7%) compared to the partial form (3.3%). Our values are significantly lower than those found by Adisen in 2016 and Bayrakdar in 2017 [12,24]; such discrepancy may be due to the small sample size compared to Adisen’s study (1246 lateral Cephalograms with an age range of 10–39 years) [12]. Bayrakdar used Cone Beam Computed Tomography with a three-dimensional reconstruction for PP identification, allowing for a clear visualization of the cervical region and the subsequent possibility of distinguishing the bilateral from the mono-lateral form; the therein reported prevalence is 36.6% [24].

In reference to the frequency of PP by gender, the literature data are still discordant, although a male predilection is generally reported [12,17,24]. A study published in 2018 and performed on 734 patients demonstrated PP occurrence in 93 patients with male predominance of 13.7% compared to female (11.5%) [9]. Gibelli’s study reported a predominance in males, with a prevalence of 19.9%, in contrast to 14.6% in females [10]. Other authors disagree with such data insisting on female predominance; in fact, in 2010 Sharma highlighted a PP frequency of 5.3% in women compared to 3.7% in men [5]; our study demonstrated higher prevalence in the female gender (9.63%) than in the male gender (5.97%), although there is no statistically significant difference.

Similarly, differences in PP prevalence in terms of age are controversial, which does not allow us to understand its true etiology. Some studies reported that the complete form of PP has the highest prevalence between 15 and 18 years of age, while the occurrence of the partial form proves to be much more variable, thus supporting the etiological hypothesis of progressive calcification of PP with complete mineralization in young adults. Gibelli confirmed this theory by reporting high prevalence of PP in orthodontic patients aged 15–18 [10]. Moreover, 58% of PP in our study refers to the second age group (12–18) and 42% to the first one (4–11 years), probably in accordance with to the progressive calcification theory. Lo Giudice attempted to clarify the hypothetical progressive calcification in a longitudinal study, concluding that PP occurs in childhood but advancement in age is not related to increasing the calcification; the evaluated sample in the study was divided into three age groups (0–6 years; 7–13 years and >14 years), where the 7–13 years old group remains the most numerous one with a PP prevalence of 50%.

Recent studies focused on a possible connection between PP and dental anomalies. In 2018, Putrino analyzed 350 lateral cephalometric radiographs and OPT demonstrating PP occurrence in 66.6% of the subjects with teeth agenesis, thus suggesting that alterations in the physiological craniofacial growth process underlie both disorders [17]. The OPT analysis in our study led to the recognition of dental agenesis in 2% of cases (3/150). Among these subjects, only one presented with a complete form of PP. The statistical analysis showed no correlation between the two anomalies, in disagreement with the above-mentioned study.

Leonardi et al. compared a group of subjects with canine retention with a control group, the former exhibiting PP in 34.2% of patients compared to 15.8% in the latter [19]. Subsequently, Ghadimi et al. discovered that PP was frequently associated with canine impaction (42.9%) [13]. Both studies suggested that such a connection is possibly related to an embryological alteration of the neural crest cells from which the maxillofacial complex, as well as the cervical part of the spine, originate [13,19]. In contrast to these studies, our analysis reported that subjects with canine retention were 10.66% (16/150); among these, only 3 patients (2%) had PP. Based on these findings, the two conditions would appear unrelated.

Previous studies found an association between malformations of the upper cervical vertebrae and jaw malformation, craniofacial morphology and malocclusion [25], thus suggesting that vertebral anomaly may affect craniofacial growth leading to a non-physiological occlusion. Bayrakdar demonstrated a significant difference in the presence of PP in the three skeletal malocclusion classes, with a greater frequency in class III patients [24]; moreover, in accordance with the literature, the author listed PP among the vertebral anomalies that have a correlation with altered skeletal relationships of the maxilla [25]. Lo Giudice et al. evaluated the relationship between the frequency and type of PP as opposed to gender, age, skeletal maturity and type of skeletal malocclusion in their retrospective longitudinal study, proving a higher frequency in class II patients (14.4%), yet without a statistical significance [9]. All of these authors suggested that the basilar part of the occipital bone initiate a connection between the cervical vertebrae and the maxillary bones, thus transferring the positional alterations between them, because of the anatomical proximity. [9,24]

The group of orthodontic patients with PP of this study most frequently presented with skeletal class I, followed by class II, and eventually class III. A comparison with patients without PP revealed no significant difference, contrary to the previously cited study.

Up to the present time, no previous study described the cephalometric values of patients with PP. The authors of the current paper initially examined the values of intermaxillary angle and craniomandibular angle of the 12 subjects with PP. Our analysis demonstrated that half of the subjects belonged to the hyperdivergent and open skeletal group, and none of them presented with hypo-divergence and a skeletal deep bite.

It is widely known that the presence of PP could cause severe clinical complications such as migraine with or without aura, chronic tension headache and neck pain [11], also referred to as ‘orofacial pain’ in temporomandibular joint disorder patients [26]. Hence, several authors have examined the connection between these painful disorders and the presence of PP, mainly reporting pain symptoms in the history of the disease and the visual analogue pain scale (VAS) [27]. Such data are missing in the current study as patients were retrospectively analyzed. Moreover, the limitations of our study is the two-dimensional analysis, compared to the CBCT data implemented by Bayrackdar and Sharma [5,24]. Overlapping of bone structures on radiographs solely provides information on the presence or absence of PP, and no further information on whether it is symmetrical or bilateral. In addition, the retrospective modality did not allow for the evaluation of a possible PP evolution over time in the same patient to support or disprove the etiological hypothesis of progressive calcification.

Further studies with a larger sample size and a prolonged follow up are needed to confirm or disprove the current findings, and to verify possible differences regarding gender, age, skeletal classes, dental abnormalities, as well as hopefully its etiology.

## 5. Conclusions

PP is a common anatomical variant of C1, easily discoverable in lateral Cephalograms of the skull administered for orthodontic purposes. Therefore, the orthodontist has the ethical responsibility to diagnose PP and inform the patient about their physical condition and possible subsequent clinical complications. Furthermore, the presence of PP could initiate taking therapeutic measures and executing follow-up programs for dental and non-dental clinical correlations. No statistical correlation between the presence of PP and age, sex, canine retention and skeletal malocclusions was discovered, thus the authors suggested performing further perspective studies with 3D imaging techniques to evaluate the possible relationships between PP and predicted jaw growth.

## Figures and Tables

**Figure 1 healthcare-10-01234-f001:**
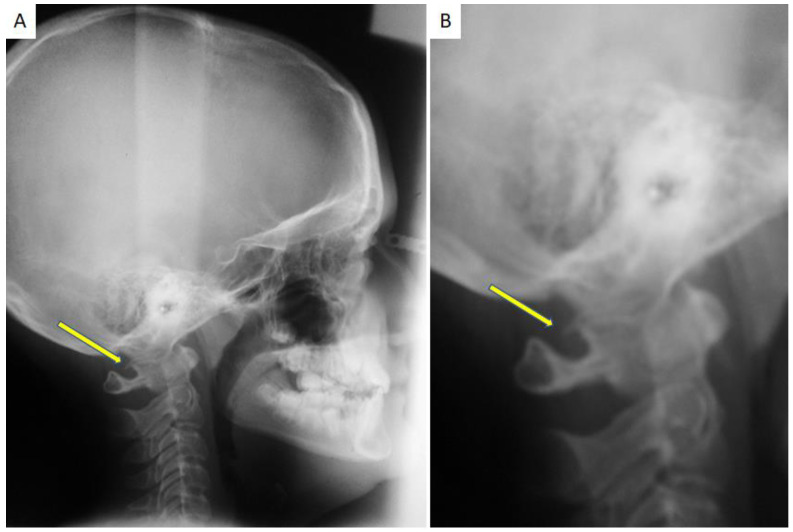
(**A**) A lateral Cephalogram of the skull performed for an orthodontic purpose occasionally showing the presence of PP; (**B**) the same radiogram with a magnification that highlights the incomplete calcification of the PP.

**Figure 2 healthcare-10-01234-f002:**
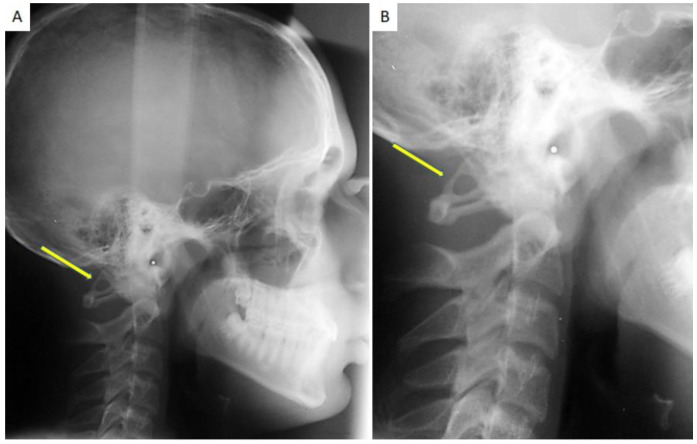
(**A**) A lateral Cephalogram of the skull performed for an orthodontic purpose occasionally showing the presence of PP; (**B**) the same radiogram with a magnification that highlights the complete calcification of the PP.

**Table 1 healthcare-10-01234-t001:** Samples description.

	Male	Female	Total
Patients	67 (45%)	83 (55%)	150
Age	12.8	11.6	10.3 (±2.7)
PP	4 (2.7%)	8 (5.3%)	12 (8%)
Dental Agenesis	1 (0.7%)	2 (1.3%)	3 (2%)
Impacted canine	7 (4.7%)	9 (6%)	16 (10.7%)

**Table 2 healthcare-10-01234-t002:** Frequency distribution of PP types by gender.

		Type of PP		
	Absence of PP	Partial PP	Complete PP	Total
Male	63	1	3	67
Female	75	4	4	83
Total	138	5	7	150
χ^2^: 1.2944; *p*-value: 0.5235				

**Table 3 healthcare-10-01234-t003:** Frequency distribution of type of PP among different age groups.

		Type of PP		
	Absence of PP	Partial	Complete	Total
From 4 to 11 years	78	3	4	85
From 12 to 18 years	60	2	3	65
	138	5	7	150
χ^2^: 0.0245; *p*-value: 0.987849				

**Table 4 healthcare-10-01234-t004:** Frequency distribution of the type of PP among different groups of dental agenesis.

	With PP	Without PP	Total
Patients with dental agenesis	1	2	3
Patients without dental agenesis	11	136	147
Total	12	138	150
χ^2^ with Yates correction: 0.3124; *p*-value: 0.5762			

**Table 5 healthcare-10-01234-t005:** Frequency distribution of the type of PP among different groups of dental inclusion.

	With PP	Without PP	Total
Patients with palatally impacted canines	3	13	16
Patients without palatally impacted canines	9	125	134
Total	12	138	150
χ^2^ with Yates’s correction: 1.4148; *p*-value: 0.2342			

**Table 6 healthcare-10-01234-t006:** Frequency distribution of type of PP among different groups of malocclusion classes.

		Type of PP		
	Absence	Partial	Complete	Total
Class I	41	3	3	47
Class II	53	1	3	57
Class III	44	1	1	46
Total	138	5	7	150
χ^2^: 3.0645; *p*-value: 0.547086				

## Data Availability

Not applicable.

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
