# Peer review of "Calcification of the Atlanto-Occipital Ligament (Ponticulus Posticus) in Orthodontic Patients: A Retrospective Study"

_healthcare, 2022, doi:10.3390/healthcare10071234_

Round 1
Reviewer 1 Report
Comments and Suggestions for Authors: The current study addresses an interesting topic in the field of Orthodontics. Study design is generally well developed, but needs improvement; while English needs a major revision to come out a nice paper at the end.
- The referee suggests improving the title, for example: Calcification of the Atlanto-Occipital Ligament (Ponticulus Posticus) in Orthodontic Patients: A Retrospective Study
- The abstract must be revised after major corrections to the manuscript. In line 25, express the percentages (4.7% and 3.3%) with dots. This is also noted in lines 57-58 and in Table 1.
- Why did the authors mention two terms "lateral cephalograms" and "tele-radiographies", where the meaning is the same? This is also noted when reading the entire manuscript. You have to choose one that you want to keep from the beginning to the end of the paper.
- The introduction section should be revised by correcting it in the right way. In line 46, “plain radiograms” is not clear here as a notion. Please, lowercase these words: “Literature (line 49)”; “Arcuate and Foramen (line 50)”. Perhaps it sounds more scientifically "mood" instead of "aura" in line 66 and other places where it is used as a terminology (line 267). The sentence in lines 74-76 creates confusion and why is the "Scientific Community" mentioned here?! The referee suggests putting the Italian state next to the University of Bari on line 84 as well; the same for line 18 in the abstract section.
- The Materials and Methods section does not clearly reflect the patients taken in the study, from Southern Italy. When it comes to Southern Italy, you have to take a big size from all the cities of the respected regions, which are many. The importance of these southern patients is not shown in any of the lines, even in the aim paragraph, being emphasized in the title. This is why it is better to delete from the title "Southern Italy", because creates confusion.
In line 90 it is better to put the number or protocol of the Ethics Document taken for the current study. In lines 97 and 104 it must be the same terminology for the legend of the respective figure. A suggestion: move both figures below line 115.In line 117 put the full names of the angles.
- Tables should be formatted the same way according to the journal's guidelines. On line 165, it is not correct in English to begin a new sentence with a number; please review the sentence. While, in table 5 the fourth column is not complete.
- In line 190 it must be America. Southern Italy is generally used in line 194, better review the message of this work. The sentence in lines 284-286 is not clear; perhaps the authors should revise their future perspectives.
- Please capitalize “southern” in line 288! The last sentence of the conclusions (lines 293-297) creates confusion.
Author Response
Every suggested correction was accepted by the authors who modified the paper text.
Southern Italy indicated the only geographical location of Bari: it doesn't mean that the study was conducted on the whole southern Italy orthodontic patients. To avoid confusion as rightly reported by you, this sentence was deleted.
Ethical committee is not needed because it's a retrospective study conducted on lateral cephalograms, performed to get the correct diagnosis and set the best orthodontic treatment plan individualized on each patient, regardless of the research of Ponticulus Posticus, which was researched at a later time.
Thank you for your suggestions that improved our paper.
Reviewer 2 Report
1.overall language and grammar of the article has to be improved
2.line 21: meaning? patients with congenital syndromes were later excluded from the study?
3. sample size calculation and ethical approval is required
4. line 136: must be mentioned in the methodology
5. line 129: it must be 11-18?
otherwise what happened to children from 11-12?
rest of the comments are given in the file attached below

Author Response
2. All patients with congenital syndromes were excluded from the study
3. Ethical approval is not needed because this is a retrospective study, conducted on lateral cephalograms, performed to get the correct diagnosis and set the best orthodontic treatment plan individualized on each patients, regardless the research of Ponticulus Posticus, which was researched at a later time.
5. 11 years old patients were included into first group, 12 years old patients were included into second group.
Paper text was modified following your suggestion.
Thank you to improving our work.
Round 2
Reviewer 1 Report
The authors improved their manuscript! But in lines 138 and 171, both sentences must be correct, because they start with numbers.
Author Response
Lines 138 and 171 were corrected with proper sentence disposition.
English form was improved
Thank you for remind.